# Unveiling a flip-over retention mechanism in the gas-phase $Cl^-$ + $(CH_3)_3CI$ $S_N2$ reaction

Xiaoxiao Lu [1,2], Jennifer Meyer [3,4], Lulu Li[5], Eduardo Carrascosa[3], Björn Bastian [3], Tim Michaelsen[3], Bina Fu [2,6,7] ✉, Dong H. Zhang [2,6,7] & Roland Wester [3] ✉

Bimolecular nucleophilic substitution ($S_N2$) and base-induced elimination (E2) reactions play crucial roles in chemistry. Here, we report a combined study of ion-molecule crossed-beam 3D velocity map imaging experiments and dynamics simulations on an accurate 39-dimensional potential energy surface for the $Cl^-$ + $(CH_3)_3CI$ reaction, allowing the most rigorous investigation of the atomistic dynamics. Good agreement between experimental and theoretical product angular and energy distributions is achieved, revealing that predominantly direct E2 reactions produce the majority of highly excited neutral products and slow ion product distributions. Moreover, we uncover a direct "flip-over" retention mechanism for the $S_N2$ reaction, where substitution occurs via the direct flipping of the tert-butyl group, leading to retention of the tetrahedral carbon center. This mechanism manifests as a predominant forward-scattering feature in the product angular distribution and provides an important perspective on substitution dynamics in organic reactions.

Bimolecular nucleophilic substitution ($S_N2$) and base-induced elimination (E2) reactions are some of the most fundamental reaction mechanisms in physical organic chemistry[1–3]. In the realm of practical chemical reactions, the interplay between solvent effects and intrinsic reaction dynamics significantly shapes reactivity. Micro-solvation studies, focusing on a few solvent molecules, have offered a bottom-up approach to investigate the roles of the solvent and understand multidimensional reaction dynamics[4–6]. On short timescales, the direct atomic-level dynamics of chemical reactions can play a dominant role in solution-phase conditions[7,8]. Gas-phase investigations, free from solvation effects, offer valuable insights into organic reaction mechanisms and the underlying factors governing these reactions.

In a fundamental $X^-$ + RY gas-phase reaction, where a nucleophile attacks an alkyl halide, the simplest methyl halides exclusively follow the $S_N2$ pathway[9–15]. A prototypical $S_N2$ reaction is commonly believed to proceed through the Walden inversion mechanism, involving a back-side attack. In this process, a nucleophile approaches the alkyl halide from one side of the α-carbon, substitutes a leaving group on the opposite side, and leading to the inversion of the tetrahedral carbon center[16]. A crucial characteristic of this mechanism lies in its stereo-specificity. This implies that an inversion invariably takes place, leading to a distinct configuration of the product molecule that is the inverse of the reactant's configuration. The concept is important because even molecules with identical atomic constituents can exhibit drastically different chemical and physical properties based solely on their spatial orientation. Hence, stereospecificity is essential in organic chemistry for the synthesis of complex molecules. The Walden inversion mechanism serves as the most common representation of the stereospecificity in the $S_N2$ mechanism. Research on the simplest model reaction, $X^-$ + $CH_3Y$, has unveiled numerous intriguing mechanisms, expanding our understanding of $S_N2$ beyond the

[1]Interdisciplinary Research Center for Biology and Chemistry, Liaoning Normal University, Dalian, China. [2]State Key Laboratory of Chemical Reaction Dynamics, Dalian Institute of Chemical Physics, Chinese Academy of Sciences, Dalian, China. [3]Institut für Ionenphysik und Angewandte Physik, Universität Innsbruck, Technikerstraße 25, Innsbruck, Austria. [4]Fachbereich Chemie und Forschungszentrum OPTIMAS, RPTU Kaiserslautern-Landau, Erwin-Schrödinger Straße 52, Kaiserslautern, Germany. [5]National Engineering Research Center of Lower-Carbon Catalysis Technology, Dalian Institute of Chemical Physics, Chinese Academy of Sciences, Dalian, China. [6]University of Chinese Academy of Sciences, Beijing, China. [7]Hefei National Laboratory, Hefei, China. ✉e-mail: bina@dicp.ac.cn; roland.wester@uibk.ac.at

traditional Walden inversion picture suggested in organic chemistry textbooks[11,17–19]. Notable examples include the abstraction-induced double-inversion mechanism initially observed in F⁻ + CH₃Cl[20], the roundabout pathway involving an initial CH₃Y rotation identified in Cl⁻ + CH₃I[21], and the front-side attack mechanism[12,22,23]. In contrast to the Walden inversion mechanism, both double inversion and front-side attack provide new pathways for retention of configuration in S$_N$2 reactions, thereby introducing another type of stereospecificity. Similar retention-like stereochemical behavior, namely torsion mechanism, was reported in S$_N$2-like hydrogen substitution reactions such as H/Cl + SiH₄[24], further highlighting the diversity of inversion and retention dynamics in substitution processes.

With the increase in the degree of methylation from a methyl residue to ethyl, isopropyl, and tert-butyl, the E2 pathway emerges as a competing channel. Understanding the intricacies of these reactions is essential for predicting and controlling the outcomes in organic synthesis. In gas-phase physical organic chemistry, much experimental[25–28] and theoretical effort[29–34] has been devoted to the investigation of this competition. Besides the influence of nucleophile[34–36] or leaving groups[37], steric effects[34,38–40] are also hot topics for research.

In our previous work, experimental findings revealed that the S$_N$2 reactivity in the F⁻ + (CH₃)₃CI reaction is extremely small compared to E2[41], an observation further substantiated by dynamics simulations based on a full-dimensional potential energy surface (PES)[38]. Intriguingly, the nearly absence of S$_N$2 reactivity is not attributed to the steric hindrance of the (CH₃)₃C bulk but rather to the high E2 reactivity of the bulk. The "intrinsic" reactivity of S$_N$2 was found to be comparable to E2 when theoretically obstructing the E2 pathway. It is noteworthy that F⁻ exhibits pronounced strength as a Lewis base, whether functioning as a nucleophile or a protophile. Given that Cl⁻ is a weaker protophile than F⁻, we expect that the E2 mechanism involving Cl⁻ should not be as dominant. This makes the competition between E2 and S$_N$2 for Cl⁻ +

(CH₃)₃CI an intriguing question. Eq. (1) summarize the two competing S$_N$2 and E2 channels for the title reaction:

$$Cl^- + (CH_3)_3CI \begin{cases} \rightarrow (CH_3)_2CCH_2 + HCl + I^- & (E2) \\ \rightarrow (CH_3)_3CCl + I^- & (S_N2) \end{cases} \quad (1)$$

Substituting F⁻ with the more moderate Cl⁻ presents a more conducive system for exploring the interplay between S$_N$2 and E2 and associated mechanisms. The provided reasoning lays the groundwork for a collaborative, experimental and theoretical exploration of the Cl⁻ + (CH₃)₃CI reaction, aiming for the utmost comprehensiveness and precision.

In this work, we report a combined experimental and theoretical study based on ion-molecule crossed-beam 3D velocity map imaging and quasi-classical trajectory calculations on an accurate full-dimensional (39-dimensional) PES, to probe the intricate atomic-level dynamics and underlying mechanisms of the Cl⁻ + (CH₃)₃CI reaction. Good agreement is observed for the product angular and kinetic energy distributions. Our analysis shows that direct E2 reactions are predominantly responsible for the slow ion products and, correspondingly, for highly excited neutral products. Furthermore, we uncover a direct "flip-over" retention mechanism for the S$_N$2 pathway of this reaction.

## Results
### Potential energy surface
To date, a comprehensive analytical PES has been absent for the Cl⁻ + (CH₃)₃CI reaction, posing a notable gap in our understanding of this process. The development of an accurate, global, and full-dimensional PES is crucial for efficient dynamics simulations. Here, a new full-dimensional PES for Cl⁻ + (CH₃)₃CI was developed by the fundamental invariant neural network (FI-NN)[42,43] approach based on roughly 256,000 CAM-XYG3/AVTZ(-PP) energy points. Figure 1 illustrates a

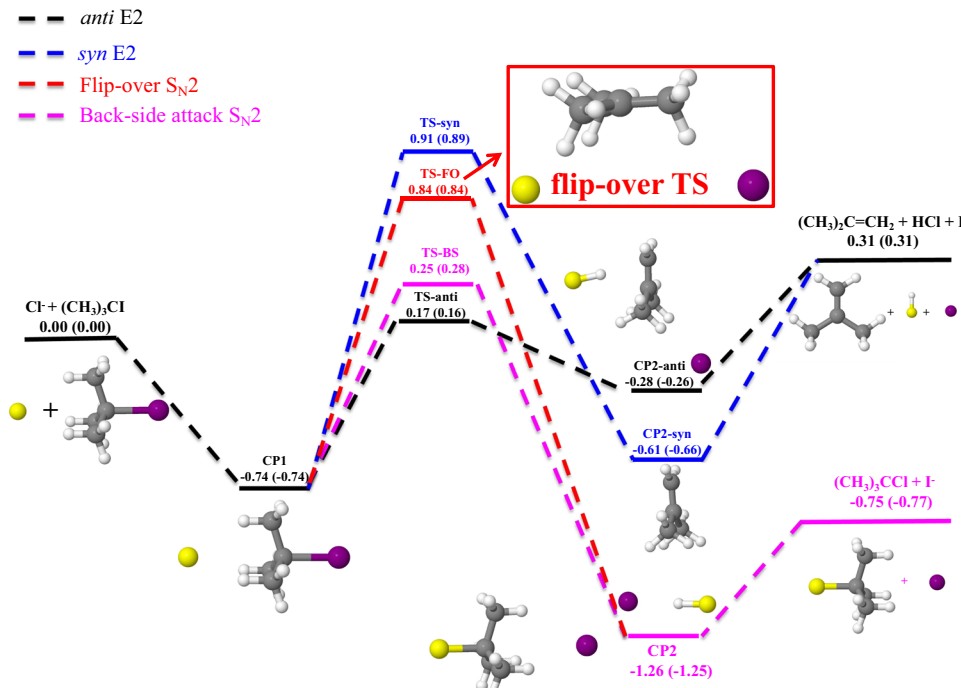

**Fig. 1 | Schematic PES of Cl⁻ + (CH₃)₃CI.** Black curve: *anti*-E2; Blue: *syn*-E2; Red: Flip-over S$_N$2; Magenta: Back-side attack S$_N$2. All the relative energies, obtained from CAM-XYG3/AVTZ(-PP) and CCSD(T)/AVTZ(-PP) methods, are given in eV (values in parenthesis refer to energies calculated with the CCSD(T) method). Source data are provided as a Source Data file.

comparison between the benchmark CCSD(T)/AVTZ(-PP) and CAM-XYG3/AVTZ(-PP) energies for all stationary points, highlighting the comparable accuracy of the latter with a reduced computational burden. The overall root-mean-square error (RMSE) for all data points is 18.3 meV. Supplementary Figs. 1–3 presents the distribution of fitting errors, optimized geometries, and contour plots, respectively.

In Fig. 1, one can see that the attacking $Cl^-$ anion and $(CH_3)_3CI$ can react through two distinct mechanisms. The *anti*-E2 pathway shows a barrier (TS-*anti*) of 0.17 eV, a post-reaction minimum (CP2-*anti*), and an endothermic energy of 0.31 eV. The back-side attack $S_N2$ pathway, known for Walden inversion, exhibits a barrier (TS-BS) differing by 0.08 eV from TS-*anti*. This pathway, however, presents a notable exothermic energy of 0.75 eV and features a deep post-reaction ion-dipole bound potential well (−1.26 eV) labeled as "CP2". Note that each channel shares a common pre-reaction minimum (CP1) with $C_{3v}$ symmetry at the entrance of the reaction, which lies −0.74 eV below the reactant asymptote. Besides the *anti* configuration, elimination reactions can also occur through the *syn* configuration, known as *syn*-E2, exhibiting a substantial barrier height of 0.91 eV and a post-reaction minimum (CP2-*syn*) at −0.61 eV. Notably, we have identified a direct $S_N2$ mechanism denoted as "flip-over", characterized by a relatively high reaction barrier of 0.84 eV (TS-FO), but lower than TS-*syn*. Two visual representations of this mechanism are provided in the Supplementary material (see Supplementary Movie 1 ($b = 6.5$ bohr, forward scattering) and Movie 2 ($b = 1.5$ bohr, backward scattering)). Additional insights into the dynamics of the unprecedented atomistic mechanism are provided below in the section "flip-over $S_N2$ mechanism".

**Reaction dynamics.** We investigated the $Cl^- + (CH_3)_3CI$ reaction dynamics using a combination of ion-molecule crossed-beam 3D velocity map imaging experiments and quasi-classical trajectory (QCT) simulations on a new accurate full-dimensional potential energy surface. Figure 2a and b display the experimental angle and energy-differential cross-sections for $I^-$ products from both E2 and $S_N2$ reactions, with $I^-$-velocity distributions presented in the center-of-mass frame at two relative collision energies ($E_{rel}$) of 1.1 eV and 1.9 eV. At $E_{rel} = 0.4$ eV the theoretical angular distribution (Supplementary Fig. 4) exhibits forward-backward symmetry, suggesting indirect dynamics with the formation of a pre-reaction complex that has a relatively long lifetime. As the collision energy increases, the angular distribution shows greater intensity in the forward scattering hemisphere, as evident in Fig. 2, indicating more direct dynamics of the reaction. In particular, the predominance of forward reactive scattering at $E_{rel} = 1.9$ eV is a clear signature of dominantly direct dynamics[34,40,41]. A comparison between theoretical angular distributions (Fig. 2c, d, red) and experimental ones (Fig. 2c, d, black) reveals a generally excellent agreement, except for small differences at 1.1 eV. The QCT-simulated cross sections of E2 and $S_N2$ pathways (Supplementary Fig. 5) as a function of collision energy together with the opacity functions (Supplementary Fig. 6) indicate the dominance of E2 in its competition with the $S_N2$ channel. The E2 pathway contributes roughly 78% and 83% to the overall reactivity at the collision energies of 1.1 eV and 1.9 eV, respectively. When the E2 pathway is completely blocked on a modified PES, $S_N2$ reactivity undergoes a substantial twofold increase. This observation suggests that in the $Cl^- + (CH_3)_3CI$ reaction, the steric hindrance effect corroborates findings from our previous exploration of the strong Lewis base $F^-$ reaction: It is not the steric hindrance that hinders the reactivity of $S_N2$, but the high reactivity of the $(CH_3)_3$ bulk leads to the predominance of the E2 reaction[38,41]. The QCT simulated results show that the direct mechanism of E2 increases with rising collision energy, as evidenced by the branching ratios of direct and indirect pathways provided in Table 1. In the QCT simulations, a soft zero-point energy constraint was applied to the E2 trajectories to ensure that the total vibrational energies of HCl and $(CH_3)_2CCH_2$ products were not lower than the sum of their respective ZPEs. All $S_N2$

products were found to satisfy the ZPE constraint, due to the large exothermicity of this channel.

Furthermore, the translational energy distributions of $I^-$ product ions are compared in Fig. 2e, f. In general, the two experimental translational energy distributions exhibit a similar trend with respect to collision energy, peaking around 0.2 eV and 0.3 eV before vanishing near 0.6 eV. Both peaks obtained from QCT calculations are slightly shifted to lower energies by approximately 0.1 eV, with the tail up to ~0.5 eV, but show overall a very good agreement with the experiment. The good agreement for the $I^-$ kinetic energy is equivalent to a good agreement between experiment and QCT simulations for the sum of the internal energies of the two neutral products and their relative recoil energy. It's worth noting that in the QCT calculations, there is a small intensity peak at 0.7 eV originating from the direct $S_N2$ reaction.

From the experiment, we cannot directly measure the energy partitioned into internal excitation of HCl and $(CH_3)_2C = CH_2$ due to the three-body break-up in the elimination process. However, the QCT simulations for the E2 reaction allow us to separate the energy partitioning between $(CH_3)_2C = CH_2$ and HCl internal excitation, as well as the relative recoil energy. Supplementary Fig. 7 displays the internal energy distributions of the product $(CH_3)_2C = CH_2$ and HCl individually at the two collision energies of 1.1 and 1.9 eV, as well as the distribution of their sum, and compares them to the distribution of the sum of the total internal energy of the products and the energy partitioned into their recoil ($E_{int} + E_{recoil}$). We can see that most energy is channeled into the internal energy of the E2 products and not their recoil energy for the $Cl^- + (CH_3)_3CI$ reaction, which is consistent with the behavior in the $F^- + (CH_3)_3CI$ reaction[38]. In contrast, most additional collision energy goes into the recoil energy in the $F^- + CH_3CH_2Cl$ reaction[44]. These results indicate that more energy is deposited into the internal energies of products, most likely due to the larger backbone and more degrees of freedom in the $(CH_3)_2C = CH_2$ product than in $CH_2 = CH_2$.

**Flip-over $S_N2$ mechanism.** With the trajectory simulations, we can accurately determine the contributions of products $I^-$ ions formed by the E2 and $S_N2$ mechanisms to the overall angular distribution of $I^-$ ions. In Fig. 3a, the angular distributions of $I^-$ ions produced via the E2 and $S_N2$ channels at the collision energy of 1.9 eV are depicted. As shown in Table 1, 89% of the E2 channel presents a direct mechanism, predominantly yielding forward scattering signals. Conversely, the total angular distribution of $I^-$ ions in the backward direction stems from the $S_N2$ reaction. As expected, the majority of $S_N2$ reactions exhibit a backward scattering signal through the direct Walden inversion mechanism, but forward scattering characteristics are also non-negligible.

We further examined those forward-scattered trajectories, among which, in addition to indirect $S_N2$ reactions, an interesting $S_N2$ reaction mechanism was identified. The most distinctive feature of this mechanism is its stereospecificity, i.e., the retention of the configuration of the central carbon in the tert-butyl group throughout the reaction. Note that trajectory analysis indicates that no front-side attack events occurred at any collision energies for this system due to its significantly high reaction barrier. We computed the dihedral angles formed between the alpha carbon ($\alpha$-C) atom and the three beta carbons ($\beta$-C) within the $S_N2$ trajectories before and after the reaction to figure out the contribution of the $S_N2$ retention mechanism. Figure 3b presents the angular distributions of Walden inversion $S_N2$ and Flip-over retention $S_N2$. The angular distribution of retention $S_N2$ reactions via the new mechanism, shown by the blue line, predominantly exhibits a forward-scattering pattern. This is attributed to large impact parameter collisions, which promotes a larger momentum of inertia for the tert-butyl group, thereby facilitating the occurrence of the flip-over $S_N2$ event, as discussed below (Supplementary Movie 1). A minor intensity in the backward direction is ascribed to direct flip-over at smaller $b$ leading to momentum reversal (Supplementary movie 2).

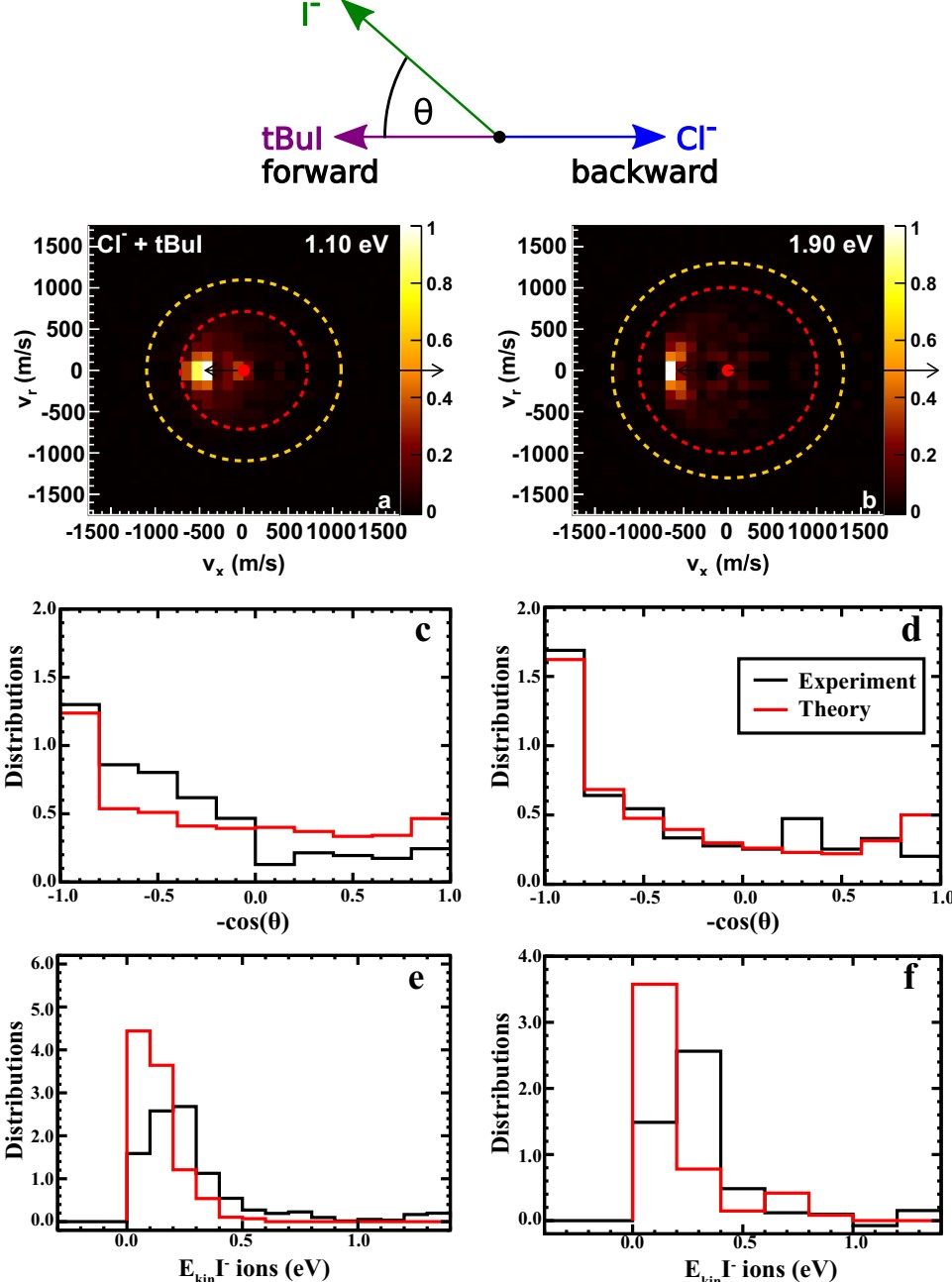

**Fig. 2 | Theory and experiment. a, b** Two-dimensional representation of the experimental 3D scattering distribution of I⁻ product ions at collision energies of 1.1 eV (**a**) and 1.9 eV (**b**). The colour scales are shown on the right side of the images. Superimposed circles represent the kinematic cut-off for $S_N2$ (orange) and E2 (red). The Newton diagram at the top illustrates the relative orientation of the velocity vectors of the reactants and the I⁻ product ions. **c–f** Comparisons of normalized angular distributions (**c, d**) and translational energy distributions (**e, f**) of product I⁻ in the center-of-mass frame at collision energies of 1.1 and 1.9 eV between experimental measurements (black curve) and QCT calculations (red curve). Experimental data at 1.1 eV is taken from ref. 34. Source data are provided as a Source Data file.

We refer to this mechanism as "flip-over" mechanism based on its unique characteristics, which is distinct from "front-side" attack and "double-inversion" configuration retention mechanisms. A typical flip-over trajectory is shown in Fig. 4b and the corresponding animation is given in the Supplementary Movie 2. Obviously, the current PES accurately reproduces the ab initio energies for the flip-over trajectory. From those time snapshots, it is evident that the Cl⁻ ion first approaches the tert-butyl iodide near the CP1 region, followed by pronounced elongation of the C–I bond. Subsequently, the Cl⁻ recoils at a short distance, followed by the flipping motion of the tert-butyl group, resembling the turning of a page in a book (rotating along the axis

perpendicular to the $C_\alpha$-I bond, as indicated by the green arrow). Finally, the Cl⁻ ion attacks the central carbon and directly displaces I⁻, leading to the retention of configuration. To clarify the sequence of structural events, Fig. 4a presents the time evolution of the Cl-C and C-I distances for a representative trajectory. The traces reveal that in the flip-over mechanism, C-I bond elongation precedes Cl-C bond formation. However, in the direct rebound mechanism, these two events occur simultaneously, and the characteristic flipping motion is absent. In addition, we also computed the ratio of the flip-over trajectories at a higher collision energy of 2.2 eV, revealing a slightly higher occurrence, approximately 10% of total $S_N2$ reactivity. The "flip-over"

**Table 1 | Branching fractions of E2 and S$_N$2 pathways and their subtypes at different collision energies**

| Collision energy (eV) | Mechanism | Fraction (%) | Subtype | Subtype fraction (%) |
|---|---|---|---|---|
| 1.1 | E2 | 77.6 | direct | 86.1 |
| | | | indirect | 13.9 |
| | S$_N$2 | 22.4 | Walden inversion | 98.9 |
| | | | Flip-over retention | 1.1 |
| 1.5 | E2 | 80.1 | direct | 87.2 |
| | | | indirect | 12.8 |
| | S$_N$2 | 19.9 | Walden inversion | 96.7 |
| | | | Flip-over retention | 3.3 |
| 1.9 | E2 | 82.6 | direct | 88.8 |
| | | | indirect | 11.2 |
| | S$_N$2 | 17.4 | Walden inversion | 93.4 |
| | | | Flip-over retention | 6.6 |
| 2.2 | E2 | 84.7 | direct | 89.6 |
| | | | indirect | 10.4 |
| | S$_N$2 | 15.3 | Walden inversion | 92.6 |
| | | | Flip-over retention | 7.4 |

Fractions are normalized to 100% for each collision energy. "Fraction (%)" denotes the proportion of reactive trajectories leading to the E2 or S$_N$2 mechanism relative to the total reactive trajectories, and "Subtype fraction (%)" indicates the relative distribution among subtypes.

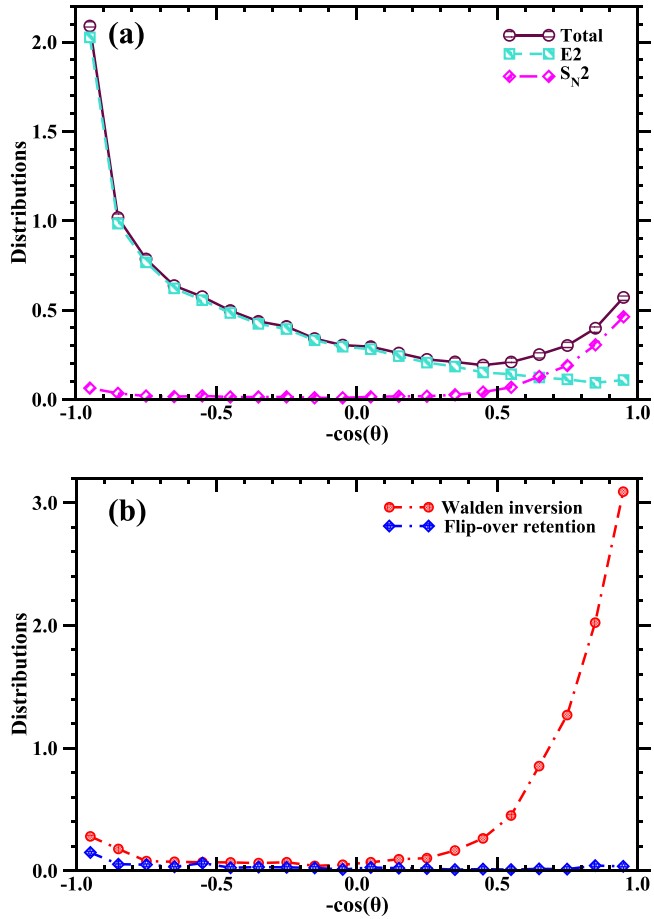

**Fig. 3 | Theoretical angular distributions. a** Angular distribution at a collision energy of 1.9 eV, showing total I$^-$ products (in brown circle) and I$^-$ resulting from E2 (in green square) and S$_N$2 (in pink diamond) reaction channels, respectively. The angular distribution area for total I$^-$ products is normalized to 1. **b** Comparison between Walden inversion S$_N$2 (in red circle) and flip-over retention S$_N$2 (in blue diamond) at a collision energy of 1.9 eV. The sum of the two S$_N$2 distributions, corresponding to total S$_N$2, is normalized to 1. Source data are provided as a Source Data file.

mechanism should be increasingly pronounced with the rise in collision energy (Table 1). To further quantify its dynamics, we analyzed the trajectory durations along this pathway, defining the reaction time as the duration time for a flip-over trajectory proceeding from the reactant side to the product side with a distance between two fragments of 12 bohr. The average duration of the flip-over trajectories is approximately 0.3 ps, consistent with a fast, direct process. Importantly, a well-defined transition state (TS-FO, as shown in Fig. 1) corresponding to the flip-over pathway has been located on the full-dimensional FI-NN PES, connecting the reactant and product valleys. This confirms that the flip-over mechanism is an energetically accessible and distinct substitution mechanism.

## Discussion

The excellent agreement between theory and experiment for the product angular and energy distributions for the Cl$^-$ + (CH$_3$)$_3$CI reaction highlights the highly accurate experimental measurements and dynamics simulations based on the full-dimensional PES. The competition between E2 and S$_N$2 for the Cl$^-$ + (CH$_3$)$_3$CI reaction is not as pronounced as in the case of F$^-$ + (CH$_3$)$_3$CI. The reactivity of S$_N$2 for the Cl$^-$ + (CH$_3$)$_3$CI reaction is about 4 to 5 times lower than that of E2, while the reactivity of S$_N$2 for F$^-$ + (CH$_3$)$_3$CI is completely suppressed. The intrinsic reactivity of S$_N$2 or E2 can only be confirmed through full-dimensional dynamics calculations based on the analytical global PES.

The unveiled flip-over mechanism in this study represents an unprecedented manifestation of S$_N$2 stereospecificity, which is distinct from the well-established front-side attack and double-inversion mechanisms. The front-side attack mechanism is a very direct process, with the nucleophile striking the C$_\alpha$-Y bond from the side of the leaving group, effecting substitution while maintaining the configuration. This typically occurs under high collision energy. In contrast,

the recently revealed double-inversion mechanism in F$^-$ + CH$_3$Cl[20] follows a two-step retention pathway via an indirect process. Both front-side attack and double inversion manifest as stereospecificity distinct from the Walden inversion in S$_N$2, leading to configuration retention. In contrast, the flip-over mechanism involves a flip of the carbon backbone and back-side attack, enriching our understanding of S$_N$2 stereospecificity. The flip-over mechanism should also be distinguished from the roundabout mechanism found for the Cl$^-$ + CH$_3$I reaction[21]. Indeed, both mechanisms are collision-induced and involve rotational motion of the alkyl group, but their detailed dynamical processes and stereochemical outcomes are fundamentally different. In the roundabout process, Cl$^-$ first strikes the side of the CH$_3$ group, causing it to rotate around the heavy I atom before attacking the carbon backside to displace I$^-$ with inversion of configuration. In the present flip-over mechanism, however, Cl$^-$ first approaches tert-butyl iodide near the CP1 region, followed by a page-turning-like flip of the tert-butyl group prior to the substitution. This collision-induced process leads to overall retention of configuration and represents a distinct dynamic pathway from the roundabout mechanism. The flip-over mechanism introduces previously unexplored dynamics, showcasing how the flexibility of the carbon backbone enables a unique stereospecificity, further diversifying the landscape of S$_N$2 substitution processes.

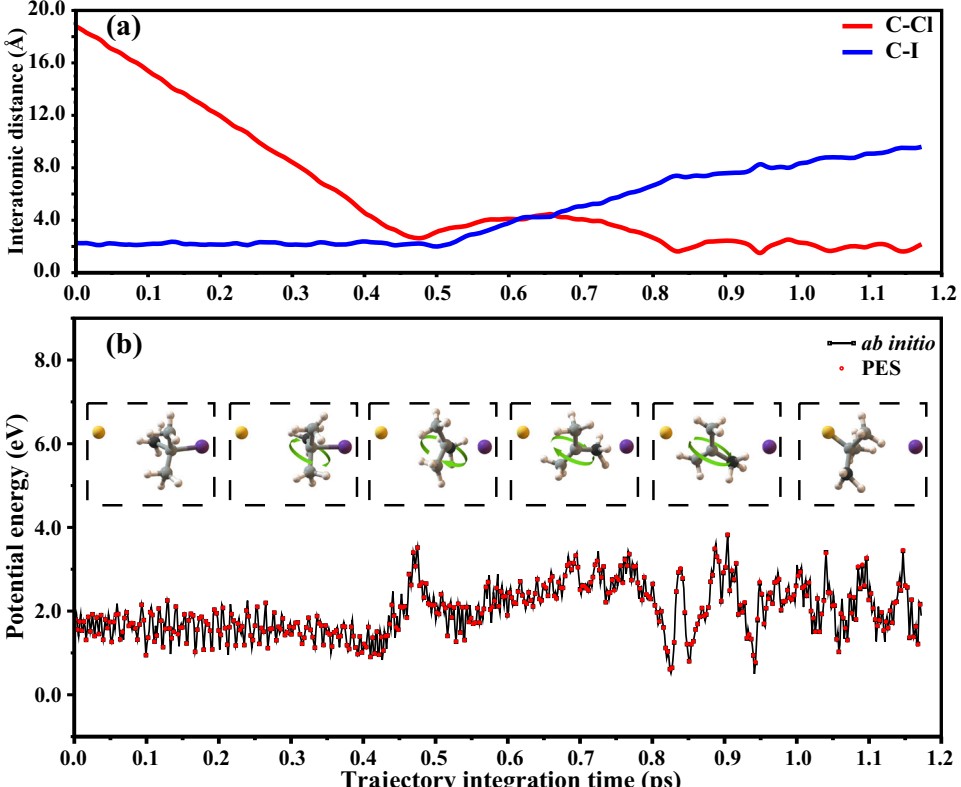

**Fig. 4 | Flip-over mechanism. a** Time evolution of the C-I (in blue curve) and C-Cl (in red curve) interatomic distances for the representative flip-over trajectory. **b** Snapshots of a representative trajectory showing the flip-over mechanism of the $Cl^- + (CH_3)_3CI$ reaction at a collision energy of 1.9 eV. The potential energies (black squares) obtained from the FI-NN PES and their corresponding ab initio energies (red spheres) are displayed as a function of integration time. A single β-carbon

atom in the tert-butyl group is labeled black, while the other carbon atoms are gray to facilitate tracking the flipping process of the tert-butyl group during the $S_N2$ reaction. All potential energies are referenced to the $(Cl^- + (CH_3)_3CI)$ reactant asymptotes, and the reactant zero-point energy (3.364 eV) was included in the QCT simulations. Source data are provided as a Source Data file.

In this study, we report a combined experimental and theoretical study for the 15-atom ion-molecule reaction $Cl^- + (CH_3)_3CI$ involving both $S_N2$ and E2 channels. The combination of ion-molecule crossed-beam velocity map imaging experiments and dynamics simulations based on an unprecedented full-dimensional (39D) PES enables a detailed investigation of reaction dynamics, including energy-dependent product-state distributions, branching ratios for $S_N2$ and E2 reactions, and unprecedented dynamical mechanisms.

For the $Cl^- + (CH_3)_3CI$ reaction, both experimental and theoretical results emphasize the dominant role of direct E2 reactions. As the collision energy increases, E2 reactions become increasingly direct, resulting in more pronounced forward scattering signals. Moreover, theoretical simulations have identified an unprecedented flip-over mechanism, as a new manifestation of stereospecificity in the $S_N2$ mechanism. Unlike the front-side attack and double-inversion mechanisms, the flip-over mechanism involves a back-side attack after a flip of the tert-butyl group. This unique process results in the retention of configuration at the reaction center. Larger impact parameters favor the flip of the carbon backbone, making forward scattering a distinctive signature of the flip-over mechanism. Unlike the previously reported roundabout process, which proceeds through a rotational motion of the alkyl group around the halogen before backside attack, the flip-over mechanism involves a direct collision-induced flipping of the carbon framework, highlighting its fundamental difference from the inversion-type roundabout mechanism.

This unprecedented configuration-retention mechanism plays an important role in the Cl- + $(CH_3)_3CI$ $S_N2$ reaction, enriching our understanding of the stereospecificity inherent to the $S_N2$ mechanism.

Investigating the involvement of the flip-over mechanism in more complex ion-molecule reactions is poised to be an intriguing path for future research.

## Methods
### Experiment
The experiments were carried out using an ion-molecule crossed beam setup combined with a velocity map imaging (VMI) spectrometer operated at 20 Hz repetition rate[33]. The $Cl^-$ ions were produced by dissociative electron attachment of the given precursor in a plasma discharge source, generated from mixtures of 5% $CH_3Cl$ in argon. After time-of-flight mass selection, the ions were guided into an octupole radiofrequency ion trap. where their kinetic energy spread is minimized through collisions with pulsed $N_2$ buffer gas. The extracted ion beam is decelerated and crossed with a supersonic neutral molecular beam at a relative collision angle of 60°. The neutral molecular beam consists of a low concentration of reactant gas seeded in helium.

Ionic products of ion-molecule reactions in the crossing region of the two beams are extracted normal to the scattering plane by pulsing on the electrodes of the VMI spectrometer. The ion position is measured with an imaging microchannel plate equipped with a phosphor screen and viewed by a CCD camera, while the arrival time is measured with a photomultiplier tube. From the ions' impact positions and arrival times, the three-dimensional velocity vectors of the reaction products are obtained. From these, the differential cross-section images are derived as a function of the velocity component parallel ($v_x$) and perpendicular ($v_r$) to the reactant's relative velocity axis. Each entry is weighted by $1/v_r$ to create an image corresponding to a slice through the three-dimensional scattering distribution.

## Theory

The current 39-dimensional PES of the title reaction was developed by fitting ~256,000 energy points using the recently proposed fundamental invariant neural network (FI-NN) method, with both potential energies and forces incorporated in the full-dimensional analytical PES[42,43]. We employed space partitioning and energy splitting methods to overcome the significant challenge of constructing a 15-atom multichannel reaction with long-range interactions in the asymptotes. The CAM-XYG3/AVTZ(-PP) method was chosen for electronic structure calculations, which is able to describe the investigated reaction well. The overall RMSE is 18.3 meV, indicating the FI-NN PES is very accurate for dynamics simulations. The PES developed and associated data points in this work are deposited in the Zonodo database.

In the QCT calculations, standard normal mode sampling was used to prepare the vibrational ground state of $(CH_3)_3CI$, and the rotational angular momentum was set to zero. The initial distance between the center of mass of $Cl^-$ and $(CH_3)_3CI$ was defined as $(x^2 + b^2)^{1/2}$, where $x$ was fixed at 38.0 bohr, and $b$ represents the impact parameter. The maximum impact parameter ($b_{max}$) is set to 10.0 bohr and 9.0 bohr for the two collision energies of 1.1 and 1.9 eV, respectively. We scanned $b$ from 0 to $b_{max}$ with a step size of 0.5 bohr. At each $b$ value, a total of 200,000 and 250,000 trajectories were simulated for 1.1 eV and 1.9 eV, to ensure robust statistical results. Further details on QCT calculations are provided in the Supplementary Information.

## Data availability

All the data generated in this study have been deposited in the zenodo database under zenodo.org (https://doi.org/10.5281/zenodo.17865514). Data supporting the findings of this manuscript are also available from the corresponding author upon request. Source data are provided with this paper.

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

## Acknowledgements

The theoretical work was supported by the National Natural Science Foundation of China (grants 22525304 (B.F.), 22288201(D.Z.), 22403090 (X.L)), the Strategic Priority Research Program of the Chinese Academy of Sciences (XDB0970203(B.F.)), the Innovation Program for Quantum Science and Technology (2021ZD0303305(D.Z.)), the robotic AI-Scientist platform of the Chinese Academy of Sciences(B.F.), Liaoning Provincial Natural Science Foundation Doctoral Research Startup Fund project (2025- BS-0768), and the Doctoral Research Start-up Program of Liaoning Normal University (2025BSL013). This experimental work was supported by the Austrian Science Fund (FWF) (R.W.), project P25956-N20. E.C. was supported by a DOC-fellowship of the Austrian Academy of Science.

## Author contributions

B.F., D.Z. and R.W. conceived and supervised the research; X.L. and L.L. performed the theoretical calculations; X.L., L.L., D.Z. and B.F. discussed and analyzed the theoretical data; and E.C. and R.W. designed the experiment. E.C., J.M. and T.M. conducted the experiments; J.M., E.C., B.B. and R.W. analyzed and discussed the experimental data; X.L., J.M., B.F. and R.W. prepared the manuscript.

## Competing interests

The authors declare no competing interests.
