## [Transparent Peer Review file · Nature Communications]

Unveiling a flip-over retention mechanism in the gas-phase $\text{Cl}^- + (\text{CH}_3)_3\text{CI}$ $\text{S}_{\text{N}}2$ reaction

Corresponding Author: Professor Roland Wester

Version 0:

Reviewer comments:

Reviewer #1

(Remarks to the Author)

Bimolecular nucleophilic substitution ($\text{S}_{\text{N}}2$) and base-induced elimination ($\text{E}2$) reactions are important in chemistry and their mechanisms have been widely recognized in text books. Ample organic syntheses utilized their stereo-specific characteristic to produce many useful products. However, Lu et al. reported a novel “flip-over” (~ 10% contribution to the total) mechanism for the $\text{S}_{\text{N}}2$ reaction channel in the typical $\text{Cl}^- + (\text{CH}_3)_3\text{CI}$ reaction, according to the ion-molecule crossed-beam 3D velocity map imaging experiment and quasi-classical trajectory reaction dynamic simulations on a reliable 39-dimensional potential energy surface (PES) fitted to 256,000 CAM-XYG3/AVTZ(-PP) calculated configurations: the substitution follows the direct flipping of the tert-butyl group. The PES was fitted by the fundamental invariant neural network (FI-NN) method that was developed by the Zhang group. Experiment and calculation are quantitatively consistent with each other for product angular and energy distributions, and both show that direct $\text{E}2$ channel is predominant with highly excited neutral products and slow ion product. Theoretical analysis showed that this new mechanism for $\text{S}_{\text{N}}2$ was manifested by a predominant forward-scattering peak in the product angular distribution. This work thus provides a new mechanism to the conventional substitution reactions, and can be widely important in organic chemistry.

Both experiment and theory are state-of-the-art. The work is overall well organized and clearly written. Congratulations to the authors for their beautiful work and continuing efforts for detailed dynamic studies on ion-molecule reactions. Therefore, I recommend to accept the manuscript once the following minor issues are addressed.

(1) In SI, refs should be cited for “we selected the recently proposed hybrid functional CAM-XYG3 developed by Xu and coworkers”.

(2) The fitting error for the interaction region is quite big, 21.3 meV, which is rather larger than for other regions, less than 5 meV. Did the authors try to reduce it? Is it because the calculated energies are of large uncertainty or are not smooth for some regions? Particularly if one notices that the fitting error increases over 0.1 eV for points with energy higher than 2 eV, as shown in SI Fig. 1.

(3) In SI, Fig. 3 was obtained using full-dimensional optimization. Is it true? I guess at least two dimensions are fixed to obtain such a contour.

(4) “the retention of the tetrahedral carbon center” is the main feature of the novel flip-over $\text{S}_{\text{N}}2$ mechanism. However, the three methyl groups are identical. With or without flipping over, the products can't be distinguished from only the experiment. It might be useful to introduce the chirality to the tetrahedral carbon center. Namely, different groups are linked to the tetrahedral carbon center. Can the experiment introduce like isotopes to do this? It should be straightforward to introduce isotopes in the simulation.

(5) The PES and the data points should be provided as supporting material as they are key products of the work.

(6) The retention mechanism was also reported in $\text{S}_{\text{N}}2$ like reaction involved in SiH_4 , such as $\text{H} / \text{Cl} + \text{SiH}_4$, which are suggested to mention to widen the scope of focus.

Reviewer #2

(Remarks to the Author)

This is a fantastic work. I enjoyed reading it very much. The authors present a wonderful work with combined ion-molecule crossed-beam 3D velocity map imaging experiments and dynamics simulations on an accurate full-dimensional potential

energy surface for the $\text{Cl}^- + (\text{CH}_3)_3\text{CI}$ reaction. Interestingly, a new mechanism called flip-over was reported. This would certainly add the knowledge of $\text{S}_\text{N}2$ mechanisms in the gas phase. I recommend the acceptance of this manuscript. I have a few points to share and discuss with the authors, which the authors may consider.

Do the authors classify the “flip-over” mechanism as a direct or an indirect mechanism? It would be great to know the average duration of the flip-over trajectories.

As shown from the animation, it seems the “flip-over” mechanism follows the four steps: 1) collision of Cl^- on the back side of $(\text{CH}_3)_3\text{CI}$, 2) elongation or breaking of the C-I bond, 3) flip of the $(\text{CH}_3)_3$ -group, and 4) formation of Cl-C bond. The direct rebound mechanism lacks the third step, and steps 2 and 4 happen simultaneously, whereas for the flip-over mechanism, steps 2 and 4 happen in sequence. I would suggest the authors illustrate the mechanism in more detail and compare it to other mechanisms. Probably, adding the change of Cl-C and C-I distance in Figure 4 would help the illustration.

Since flip-over occurs more frequently at high collision energy, it is more likely to be a collision-induced mechanism. This is similar to the roundabout mechanism. Hase group reported a type of special roundabout mechanism in JPCA 2013, 117, 7162. “For two of the trajectories, reaction occurred by the roundabout mechanism, but with retention of the CH_3 stereochemistry, a quite interesting finding. As CH_3 rotates about the massive I atom, instead of attacking the back side of the C atom, OH^- attacks the front side of the C atom, without inversion of the CH_3 -group to form the CH_3OH product.” In this work, collision induced the rotation of the CH_3 group, which, in a way, assists the inversion of the CH_3 group, thus switching the back-side attack to front-side attack. Here, in this work, collision directly induced the breaking of the C-I bond, and the $(\text{CH}_3)_3$ -group proactively flipped over. Clearly, they are different. And it would be nice to mention such comparisons. In addition, is a roundabout mechanism observed in this work?

Is this mechanism possible in the $\text{Cl}^- + \text{CH}_3\text{I}$ reaction? For the $\text{Cl}^- + (\text{CH}_3)_3\text{CI}$ reaction, in step 3, the $(\text{CH}_3)_3$ -group behaves like a free radical. Similarly, CH_3 may behave like a radical and flip.

The structure of TS-FO looks more like a front-side attack transition state, as both Cl and I are on the same side of $(\text{CH}_3)_3$ -group. One can check the Cl-C-I angle. In fact, in the animation of the flip-over mechanism, this configuration (TS-FO) doesn't show up, as Cl and I atoms are on the opposite side of the $(\text{CH}_3)_3$ -group. Please double-check.

If the authors can provide more numerical values (for example, using a table) for some critical information, it will greatly increase the clarity of this work. I will name a few. The ratio of $\text{S}_\text{N}2$ and E_2 channels, the percentage of flip-over mechanism over all $\text{S}_\text{N}2$ product channels, at different collision energies, experimentally and from simulation. I may have missed the information in the main manuscript.

Figure 4: What is the reference point of the potential energy? If the separating reactants are the reference points, some points are nearly 4.0 eV; would ZPE leaking be involved?

Reviewer #3

(Remarks to the Author)

This paper describes a new study of the reaction between Cl^- and $(\text{CH}_3)_3\text{CI}$. The experimental work is conducted on a crossed molecular beam 3D VMI instrument developed over a number of years by the Innsbruck group. Angular and kinetic energy release distributions are presented at two collision energies. The angular distributions are found to be broad, but forward peaking, with forward scattering increasing with increasing collision energy. The product I^- ions are found to be produced with a wide range of kinetic energies, peaking at around 0.2 to 0.3 eV. The experiments are supported by some very impressive potential energy surface (PES) and quasi classical trajectory (QCT) calculations. The calculations generally agree reasonably well with experimental data - impressively so for a system of this size. Perhaps more importantly, the theory is used to shed further light on the reaction mechanisms at play at the collision energies studies. They suggest a strong preference for the operation of the E_2 mechanism, with a more minor role for the $\text{S}_\text{N}2$ mechanism. In addition the authors are able to demonstrate, through analysis of the PES and the trajectories, the operation of two $\text{S}_\text{N}2$ mechanisms, one the classic Walden inversion, and the other a newly identified $\text{S}_\text{N}2$ process which the authors term a flip-over mechanism.

The paper is generally clearly presented and the material of likely interest to readers of Nature Communications. The theoretical calculations, both PES and QCT aspects, are particularly impressive. Overall, in my view the paper warrants publication in Nature Communications, but I do have a number of concerns and queries that I believe the authors should address before publication.

1. The authors emphasize the relevance and importance of their work in connection to organic synthesis. Whilst of course there is a connection, it seems to me that the role of the solvent cannot be neglected, particularly for something like the flip-over mechanism, which it seems to me could be highly constrained by solvent interactions. I would also suggest that the collision energy used is unlikely to play a significant role in organic synthesis.

2. The paper does not provide any experimental evidence for the flip-over mechanism. As impressive as the calculations are, can the calculations really be trusted sufficiently to unveil this new mechanism, particularly given that it is clearly a very minor pathway?

3. As noted in 2, the flip-over mechanism plays a very minor role, even at the highest collision energy studied. Can it really be identified as a new mechanism? Trajectories for a system as complex as this are likely to behave in very complex ways - what is the criteria for labelling them as a new mechanism. Is it the presence of an accessible transition state? This seems unclear to me, and perhaps needs clearer articulation in the manuscript.

4. In the abstract, the agreement between experiment and theory is described as quantitative. I would say it was qualitative, particularly for the kinetic energy distributions - again acknowledging the impressive nature of the calculations.

5. I feel that the challenges of zero point energy violation, and the methods used to circumvent them, are insufficiently explained in the manuscript.

6. There is reference in the SI to calculations on a modified PES. It seems to have a very big effect on the cross sections for the Sn2 process, but there is little discussion of this in the text.

7. I thought that the branching ratio data shown in tables 2 and 3 of the SI would be better placed in the main text. In addition, I believe those tables should include the fraction of trajectories which involve the E2 and Sn2 mechanisms. Those numbers appear in the text at various places, and it would be clearer if everything was collected together in one table in the main text.

8. For the non-expert, I thought it would be helpful if the reaction equations (1) appeared earlier in the text, when Sn2 and E2 are first discussed.

Reviewer #4

(Remarks to the Author)

Review of manuscript titled "Unveiling a flip-over retention mechanism in the gas-phase SN2 reaction: insights from Cl⁻ + (CH₃)₃CI dynamics" by Xiaoxiao Lu, Jennifer Meyer, Lulu Li, Eduardo Carrascosa, Björn Bastian, Tim Michaelsen, Bina Fu, Dong H. Zhang and Roland Wester.

This manuscript describes theory and experiment on the reaction of Cl⁻ with (CH₃)₃CI forming I⁻ through either a nucleophilic substitution (SN2) and base-induced elimination (E2) reaction. The E2 reaction forming three fragments I⁻ + HCl + (CH₃)₂CCH₂ and the sn2 reaction forming two products (CH₃)₃CCl + I⁻. This is distinct from reactions of Cl⁻ + CH₃R (R=halogen) where only the SN2 reaction is observed. The Wester group has studied many such systems and their experimental expertise for this type of experiment is unquestioned. The reality is that only the I⁻ anion is able to be observed at it is a somewhat unreliable reporter on the details of the reaction due to the fact that it is heavy and slowly moving and the images are of limited velocity resolution. The combination of theory and experiment is quite convincing however. From the I⁻ images one learns that the I⁻ velocities are consistent with E2 elimination and the analysis indicates that 80% of the reaction proceeds through this channel. This is different from the previously studied F⁻ plus (CH₃)₃CCl system where the E2 reaction was more dominant and the SN2 channel was extremely small.

The other more interesting result is the identification of a new reaction mechanism from trajectory calculations on the potential energy surface. Theory on such a large (many atoms and electrons) system is difficult and as I am no expert in the theory I will not comment on this aspect of the manuscript. The new mechanism involves an intriguing spin flip of the t-butyl group that is reminiscent of a sort of hindered roaming of the group such that the I⁻ anion is for a moment bound to a CH₃ group exposing the central C atom to the Cl⁻ anion for reaction. Discovery of a new reaction mechanism is always of fundamental chemical interest. It is not particularly confirmed by the experiment but comes from the theory that the experiment demanded.

In conclusion, if reviewers who are more familiar with the theory feel the predictions of the theory are accurate I would highly recommend publication for this manuscript as it contains a fundamentally new reaction mechanism for a classic reaction.

Version 1:

Reviewer comments:

Reviewer #1

(Remarks to the Author)

Thanks for the improvement and clarification. I am satisfied with the revised manuscript, which addressed all my concern and happy to recommend its acceptance.

Reviewer #2

(Remarks to the Author)

I appreciated the very careful and detailed responses provided by the authors, which fully addressed the comments. I support the acceptance of the manuscript. Congratulations to the authors.

Reviewer #3

(Remarks to the Author)

In my view, the authors have paid careful attention to the referees' comments and have addressed them in detail in the revised text. I believe the paper is now ready for publication in Nature Communications without further amendment.

We would like to thank the four reviewers for the insightful comments which are very helpful for the improvement of our manuscript. We have carefully read those comments and replied to them point-by-point as follows. The revisions are indicated by blue fonts in the manuscript.

Reviewer #1 (Remarks to the Author):

Bimolecular nucleophilic substitution (SN2) and base-induced elimination (E2) reactions are important in chemistry and their mechanisms have been widely recognized in text books. Ample organic syntheses utilized their stereo-specific characteristic to produce many useful products. However, Lu et al. reported a novel “flip-over” (~ 10% contribution to the total) mechanism for the SN2 reaction channel in the typical Cl⁻ + (CH₃)₃CI reaction, according to the ion-molecule crossed-beam 3D velocity map imaging experiment and quasi-classical trajectory reaction dynamic simulations on a reliable 39-dimensional potential energy surface (PES) fitted to 256,000 CAM-XYG3/AVTZ(-PP) calculated configurations: the substitution follows the direct flipping of the tert-butyl group. The PES was fitted by the fundamental invariant neural network (FI-NN) method that was developed by the Zhang group. Experiment and calculation are quantitatively consistent with each other for product angular and energy distributions, and both show that direct E2 channel is predominant with highly excited neutral products and slow ion product. Theoretical analysis showed that this new mechanism for SN2 was manifested by a predominant forward-scattering peak in the product angular distribution. This work thus provides a new mechanism to the conventional substitution reactions, and can be widely important in organic chemistry. Both experiment and theory are state-of-the-art. The work is overall well organized and clearly written. Congratulations to the authors for their beautiful work and continuing efforts for detailed dynamic studies on ion-molecule reactions. Therefore, I recommend to accept the manuscript once the following minor issues are addressed.

(1) In SI, refs should be cited for “we selected the recently proposed hybrid functional CAM-XYG3 developed by Xu and coworkers”.

Author reply:

We thank the reviewer for the suggestion. The corresponding reference to the hybrid functional CAM-XYG3 developed by Xu and coworkers has been added.

(2) The fitting error for the interaction region is quite big, 21.3 meV, which is rather

larger than for other regions, less than 5 meV. Did the authors try to reduce it? Is it because the calculated energies are of large uncertainty or are not smooth for some regions? Particularly if one notices that the fitting error increases over 0.1 eV for points with energy higher than 2 eV, as shown in SI Fig. 1.

Author reply:

We thank the reviewer for this valuable comment. We carefully tested various neural network architectures, and the present FI-NN model offers the best overall performance. The relatively larger fitting error of 21.3 meV in the interaction region mainly originates from the intrinsic complexity of the system, which involves four competing reaction channels and a wide energy range, rather than the uncertainty in the *ab initio* data. Although the fitting error in the interaction region is larger than 5 meV, the fitting accuracy (<0.5 kcal/mol) is very good for such a 15-atom complex reaction, which guarantees the quantitatively reliable dynamics results.

For data points with energies above 2 eV, errors larger than 0.1 eV occur in only 219 points (about 0.1% of the dataset). These deviations primarily indicate the high-dimensional nature of this reaction and challenge of the fitting. These points with slightly larger fitting errors have a negligible influence on the overall quality of the PES and the subsequent dynamics results.

(3) In SI, Fig. 3 was obtained using full-dimensional optimization. Is it true? I guess at least two dimensions are fixed to obtain such a contour.

Author reply:

We thank the reviewer for this comment. Figure 3 in the Supporting Information was plotted by performing a relaxed scan along the two reactive bond lengths. At each grid point, these two bond lengths were fixed at the specified values, while all other degrees of freedom were fully optimized.

We have clarified this point in the revised manuscript and the corresponding figure caption has been updated to read:

“Supplementary Figure 3 | Contour plots. Contour plots illustrating the back-side attack S_N2 (a) and anti-E2 (b) channels on the FI-NN PES obtained from relaxed scans along the two reactive bond lengths, with all other degrees of freedom fully optimized.”

(4) “the retention of the tetrahedral carbon center” is the main feature of the novel flip-over S_N2 mechanism. However, the three methyl groups are identical. With or without flipping over, the products can’t be distinguished from only the experiment. It might be

useful to introduce the chirality to the tetrahedral carbon center. Namely, different groups are linked to the tetrahedral carbon center. Can the experiment introduce like isotopes to do this? It should be straightforward to introduce isotopes in the simulation.

Author reply:

We thank the reviewer for this interesting suggestion. Experiments with substituted molecules that introduce asymmetries between the methyl groups are in principle possible. However, such reactants, including deuterated molecules, will need to be synthesized on site with sufficient purity and quantity, as they are not readily commercially available. Therefore, this would be a task for an independent future project.

(5) The PES and the data points should be provided as supporting material as they are key products of the work.

Author reply:

We thank the reviewer for the suggestion. The PES and the *ab initio* data points have now been provided as Supplementary Data 1.

(6) The retention mechanism was also reported in SN2 like reaction involved in SiH₄, such as H / Cl + SiH₄, which are suggested to mention to widen the scope of focus.

Author reply:

We thank the reviewer for this valuable suggestion. We agree that mentioning S_N2-like reactions involving SiH₄ can broaden the scope of the discussion. But in the literature the H + SiH₄ reaction does not involve retention. Accordingly, we have added the following sentence on **Page 3**: “*Similar retention-like stereochemical behavior, namely the torsion mechanism, was reported in S_N2-like hydrogen substitution reactions such as Cl + SiH₄²⁴, further highlighting the diversity of inversion and retention dynamics in substitution processes.*”

Reviewer #2 (Remarks to the Author):

This is a fantastic work. I enjoyed reading it very much. The authors present a wonderful work with combined ion-molecule crossed-beam 3D velocity map imaging experiments and dynamics simulations on an accurate full-dimensional potential energy surface for the Cl⁻ + (CH₃)₃CI reaction. Interestingly, a new mechanism called flip-

over was reported. This would certainly add the knowledge of SN2 mechanisms in the gas phase. I recommend the acceptance of this manuscript. I have a few points to share and discuss with the authors, which the authors may consider.

Do the authors classify the “flip-over” mechanism as a direct or an indirect mechanism? It would be great to know the average duration of the flip-over trajectories.

Author reply:

We thank the reviewer for this insightful comment. The flip-over mechanism is classified as a **direct S_N2 pathway**, as stated in the *Potential energy surface* section: “Notably, we have identified a new direct S_N2 mechanism denoted as ‘flip-over’, characterized by a relatively high reaction barrier of 0.84 eV (TS-FO), but lower than TS-syn.” We acknowledge that this point was not sufficiently emphasized, so we have updated the abstract to read: “Moreover, we uncover a novel **direct** ‘flip-over’ retention mechanism for the S_N2 reaction, where the substitution follows the direct flipping of the tert-butyl group, resulting in the retention of the tetrahedral carbon center.”

The **average reaction time** of the flip-over trajectories is approximately **0.3 ps**. We have added this information at the end of the *Flip-over S_N2 mechanism* section (**page 13**): “To further quantify its dynamics, we analyzed the trajectory durations along this pathway, defining the reaction time as duration time for a flip-over trajectory proceeding from the reactant side to product side with a distance between two fragments of 12 bohr. The average duration of the flip-over trajectories is approximately 0.3 ps, consistent with a fast, direct process.”

As shown from the animation, it seems the “flip-over” mechanism follows the four steps: 1) collision of Cl⁻ on the back side of (CH₃)₃CI, 2) elongation or breaking of the C-I bond, 3) flip of the (CH₃)₃-group, and 4) formation of Cl-C bond. The direct rebound mechanism lacks the third step, and steps 2 and 4 happen simultaneously, whereas for the flip-over mechanism, steps 2 and 4 happen in sequence. I would suggest the authors illustrate the mechanism in more detail and compare it to other mechanisms. Probably, adding the change of Cl-C and C-I distance in Figure 4 would help the illustration.

Author reply:

We thank the reviewer for this insightful comment. Following the suggestion, we have added a new panel (Fig. 4a) showing the time evolution of the Cl-C and C-I distances for a representative flip-over trajectory. This plot clearly shows that the C-I bond elongation (Step 2) occurs before the Cl-C bond formation (Step 4), confirming the

sequential nature of the flip-over process. In contrast, for the direct rebound mechanism, these two events take place simultaneously, and the characteristic flipping motion (Step 3) is absent.

Corresponding explanations have been added at the end of the *Flip-over S_N2 mechanism* section (**page 13**), where we inserted the following sentences: “*To clarify the sequence of structural events, Fig. 4a presents the time evolution of the Cl-C and C-I distances for a representative trajectory. The traces reveal that in the flip-over mechanism, C-I bond elongation precedes Cl-C bond formation. However, in the direct rebound mechanism these two events occur simultaneously and the characteristic flipping motion is absent.*”.

The caption of Fig. 4 has also been updated to describe both panels.

Since flip-over occurs more frequently at high collision energy, it is more likely to be a collision-induced mechanism. This is similar to the roundabout mechanism. Hase group reported a type of special roundabout mechanism in JPCA 2013, 117, 7162. “For two of the trajectories, reaction occurred by the roundabout mechanism, but with retention of the CH₃ stereochemistry, a quite interesting finding. As CH₃ rotates about the massive I atom, instead of attacking the back side of the C atom, OH⁻ attacks the front side of the C atom, without inversion of the CH₃-group to form the CH₃OH product.” In this work, collision induced the rotation of the CH₃ group, which, in a way, assists the inversion of the CH₃ group, thus switching the back-side attack to front-side attack. Here, in this work, collision directly induced the breaking of the C-I bond, and the (CH₃)₃-group proactively flipped over. Clearly, they are different. And it would be nice to mention such comparisons. In addition, is a roundabout mechanism observed in this work?

Author reply:

We thank the reviewer for this insightful comment and for pointing out the possible connection between the flip-over and roundabout mechanisms. Indeed, both mechanisms are collision-induced and involve rotational motion of the alkyl group; however, their detailed dynamical processes and stereochemical outcomes are fundamentally different.

In the **roundabout mechanism** reported by Hase and co-workers for the X⁻ + CH₃Y reaction (J. Phys. Chem. A 2013, 117, 7162; Science 2008, 319, 183-186), the X⁻ ion first strikes the side of the CH₃ group, inducing its rotation around the more massive I atom. After one or more revolutions, X⁻ attacks the carbon center from the backside and

directly displaces I⁻, leading to inversion of configuration at the carbon center.

In contrast, the **flip-over mechanism** identified in this work proceeds via a direct collision-induced process. The Cl⁻ ion approaches tert-butyl iodide, triggering a page-turning-like flip of the tert-butyl group, followed by Cl⁻ attack on the carbon center and displacement of I⁻. This process results in retention of configuration and represents a distinct collision-induced pathway compared to the inversion-type roundabout mechanism.

We have incorporated this comparison into the *Discussion* section (**page 14**) with the following sentences: *“Indeed, both mechanisms are collision-induced and involve rotational motion of the alkyl group, but their detailed dynamical processes and stereochemical outcomes are fundamentally different. In the roundabout process, Cl⁻ first strikes the side of the CH₃ group, causing it to rotate around the heavy I atom before attacking the carbon backside to displace I⁻ with inversion of configuration. In the present flip-over mechanism, however, Cl⁻ first approaches tert-butyl iodide near the CPI region, followed by a page-turning-like flip of the tert-butyl group prior to the substitution. This collision-induced process leads to overall retention of configuration and represents a distinct dynamic pathway from the roundabout mechanism.”*

In addition, we have added a brief explanatory note in the *Conclusion* section (**page 15**) to further highlight this distinction: *“Unlike the previously reported roundabout process, which proceeds through a rotational motion of the alkyl group around the halogen before backside attack, the flip-over mechanism involves a direct collision-induced flipping of the carbon framework, highlighting its fundamental difference from the inversion-type roundabout mechanism.”*

Finally, **no roundabout trajectories were observed** in our simulations, confirming that the flip-over pathway represents a distinct collision-induced mechanism.

Is this mechanism possible in the Cl⁻ + CH₃I reaction? For the Cl⁻ + (CH₃)₃CI reaction, in step 3, the (CH₃)₃-group behaves like a free radical. Similarly, CH₃ may behave like a radical and flip.

Author reply:

We thank the reviewer for this insightful question regarding the possible existence of a flip-over mechanism in the Cl⁻ + CH₃I reaction. Since Cl⁻ + CH₃I represents a prototypical S_N2 reaction, exploring this possibility is an intriguing direction for future research. However, extensive experimental and theoretical studies of Cl⁻ + CH₃I,

including the discovery of the roundabout mechanism, have not reported a configuration-retention pathway like flip-over. Nonetheless, constructing an accurate full-dimensional PES for $\text{Cl}^- + \text{CH}_3\text{I}$ and examining the possibility of flip-over dynamics would provide a valuable extension of the work for the $\text{S}_{\text{N}}2$ reaction.

The structure of TS-FO looks more like a front-side attack transition state, as both Cl and I are on the same side of $(\text{CH}_3)_3$ -group. One can check the Cl-C-I angle. In fact, in the animation of the flip-over mechanism, this configuration (TS-FO) doesn't show up, as Cl and I atoms are on the opposite side of the $(\text{CH}_3)_3$ -group. Please double-check.

Author reply:

We thank the reviewer for this helpful comment. We have carefully rechecked the optimized TS-FO structure and its imaginary mode. The C-Cl and C-I distances are 3.58 and 4.46 Å, respectively, and the Cl-C-I angle is 106° , which clearly differs from the typical front-side attack TS (where nucleophile and leaving group lie on the same side of the carbon and form simultaneous partial bonds to the carbon). The imaginary frequency of TS-FO corresponds to the flip-over motion connecting the reactant and product valleys on our FI-NN PES, verifying that TS-FO is associated with the flip-over pathway rather than a conventional front-side attack TS.

If the authors can provide more numerical values (for example, using a table) for some critical information, it will greatly increase the clarity of this work. I will name a few. The ratio of $\text{S}_{\text{N}}2$ and E2 channels, the percentage of flip-over mechanism over all $\text{S}_{\text{N}}2$ product channels, at different collision energies, experimentally and from simulation. I may have missed the information in the main manuscript.

Author reply:

We thank the reviewer for this insightful comment. In the revised manuscript, the data that were previously shown in Supplementary Tables 2 and 3 have been combined and moved into the main text as the new **Table 1**, which summarizes both the branching ratios of E2 and $\text{S}_{\text{N}}2$ mechanisms and the percentage of flip-over mechanism over all $\text{S}_{\text{N}}2$ product channels at different collision energies.

Figure 4: What is the reference point of the potential energy? If the separating reactants are the reference points, some points are nearly 4.0 eV; would ZPE leaking be involved?

Author reply:

We thank the reviewer for this insightful comment. The potential energy is referenced to the asymptotic reactants ($\text{Cl}^- + (\text{CH}_3)_3\text{CI}$). The reactant zero-point energy (ZPE) is

3.364 eV, and with a collision energy of 1.90 eV, the total available energy amounts to about 5.26 eV. Hence, configurations approaching 4.0 eV are energetically accessible and should not indicate ZPE leakage. The transient high-energy points arise from the conversion of collisional to potential energy during interaction. In addition, we have indicated on Page 8 that all S_N2 products were found to satisfy the ZPE constraint, due to the large exothermicity of this channel. We have clarified in the caption that all energies are relative to the reactant asymptotes and that the reactant ZPE was included in the initial sampling.

Reviewer #3 (Remarks to the Author):

This paper describes a new study of the reaction between Cl⁻ and (CH₃)₃CI. The experimental work is conducted on a crossed molecular beam 3D VMI instrument developed over a number of years by the Innsbruck group. Angular and kinetic energy release distributions are presented at two collision energies. The angular distributions are found to be broad, but forward peaking, with forward scattering increasing with increasing collision energy. The product I⁻ ions are found to be produced with a wide range of kinetic energies, peaking at around 0.2 to 0.3 eV. The experiments are supported by some very impressive potential energy surface (PES) and quasi classical trajectory (QCT) calculations. The calculations generally agree reasonably well with experimental data - impressively so for a system of this size. Perhaps more importantly, the theory is used to shed further light on the reaction mechanisms at play at the collision energies studied. They suggest a strong preference for the operation of the E2 mechanism, with a more minor role for the S_N2 mechanism. In addition the authors are able to demonstrate, through analysis of the PES and the trajectories, the operation of two S_N2 mechanisms, one the classic Walden inversion, and the other a newly identified S_N2 process which the authors term a flip-over mechanism.

The paper is generally clearly presented and the material of likely interest to readers of Nature Communications. The theoretical calculations, both PES and QCT aspects, are particularly impressive. Overall, in my view the paper warrants publication in Nature Communications, but I do have a number of concerns and queries that I believe the authors should address before publication.

1. The authors emphasize the relevance and importance of their work in connection to organic synthesis. Whilst of course there is a connection, it seems to me that the role of the solvent cannot be neglected, particularly for something like the flip-over mechanism,

which it seems to me could be highly constrained by solvent interactions. I would also suggest that the collision energy used is unlikely to play a significant role in organic synthesis.

Author reply:

We thank the reviewer for this insightful comment. We fully agree that solvent effects play a crucial role in organic synthesis and can substantially influence both the energetics and accessibility of reaction pathways, including the flip-over mechanism. The aim of the present work, however, is to establish a clear atomistic picture of the intrinsic gas-phase dynamics, where the absence of solvent enables unambiguous identification of distinct mechanistic channels such as S_N2 and E2. Such gas-phase studies provide essential benchmarks for understanding the fundamental reactivity patterns that underpin condensed-phase and solution-phase chemistry.

Regarding the collision energy, we note that the employed energy range is not intended to mimic solution conditions but rather to explore how the underlying reaction dynamics evolve with increasing energy, providing deeper insight into the energy dependence and dynamical nature of these mechanisms. Future work can include the microsolvation effects of this reaction and associated flip-over mechanism.

2. The paper does not provide any experimental evidence for the flip-over mechanism. As impressive as the calculations are, can the calculations really be trusted sufficiently to unveil this new mechanism, particularly given that it is clearly a very minor pathway?

Author reply:

The reviewer is of course correct: the experiment alone does not provide evidence for the flip-over mechanism. However, the good agreement for the energy and angular distributions when comparing experiment and calculation is strong evidence for an accurate representation of the dynamics in the simulations and the underlying potential energy surface. Further factors support the robustness of our calculations:

- (1) The imaginary frequency of flip-over transition state (TS-FO) corresponds to the flip-over motion connecting the reactant and product valleys on our FI-NN PES, verifying that TS-FO is associated with the flip-over pathway.
- (2) The flip-over mechanism identified from the trajectory analysis involves structures whose energies are in excellent agreement with the *ab initio* calculations, indicating that our PES is highly accurate and that the predicted mechanism is reliable.
- (3) Although the contribution from flip-over is small, it is definitely identified in the full-dimensional dynamics simulations based on an accurate global PES.

In summary, there is clear evidence for the new flip-over mechanism.

3. As noted in 2, the flip-over mechanism plays a very minor role, even at the highest collision energy studied. Can it really be identified as a new mechanism? Trajectories for a system as complex as this are likely to behave in very complex ways - what is the criteria for labelling them as a new mechanism. Is it the presence of an accessible transition state? This seems unclear to me, and perhaps needs clearer articulation in the manuscript.

Author reply:

We thank the reviewer for this insightful comment. We fully agree that, for a complex polyatomic reaction, identifying a new mechanism requires rigorous justification. In the present study, the flip-over mechanism is recognized as a distinct S_N2 pathway based on both its unique dynamical characteristics and the presence of a well-defined transition state (TS-FO, Figure 1) on the full-dimensional FI-NN PES. Although its branching ratio is relatively small (Table 1), the mechanism can be clearly distinguishable from established S_N2 pathways such as Walden inversion, front-side attack, roundabout, or double inversion.

To clarify this point, we have added the following sentence at the end of the *Flip-over dynamics* section: *Importantly, a well-defined transition state (TS-FO, as shown in Figure 1) corresponding to the flip-over pathway has been located on the full-dimensional FI-NN PES, connecting the reactant and product valleys. This confirms that the flip-over mechanism is an energetically accessible and distinct substitution mechanism.*

4. In the abstract, the agreement between experiment and theory is described as quantitative. I would say it was qualitative, particularly for the kinetic energy distributions - again acknowledging the impressive nature of the calculations.

Author reply:

We thank the reviewer for this comment. We agree that the scattering angle and kinetic energy histograms do not agree perfectly, but the experimental and calculated distributions represent each other in great detail. To reflect this in the wording, we have changed the expression “*Quantitative agreement ...*” to “*Good agreement ...*”.

5. I feel that the challenges of zero point energy violation, and the methods used to circumvent them, are insufficiently explained in the manuscript.

Author reply:

We thank the reviewer for this valuable comment. We agree that the treatment of zero-

point energy (ZPE) in the QCT simulations could be more clearly described in the main text. In the Supplementary Information (last paragraph), it is stated that: “*Thus, the QCT analysis of the E2 reaction takes into account trajectories that include a soft ZPE constraint of products, where the total vibrational energies of the HCl and (CH₃)₂CCH₂ products are not lower than the sum of their respective zero-point energy. In contrast, no ZPE violation was found to the products of S_N2 reaction events, due to the large exothermicity of this channel. As a result, we collected all S_N2 trajectories for analysis.*”

To clarify this in the main text, we have now added the following sentence on **page 8**: “*In the QCT simulations, a soft zero-point energy constraint was applied to the E2 trajectories to ensure that the total vibrational energies of HCl and (CH₃)₂CCH₂ products were not lower than the sum of their respective ZPEs. All S_N2 products were found to satisfy the ZPE constraint, due to the large exothermicity of this channel.*”.

6. There is reference in the SI to calculations on a modified PES. It seems to have a very big effect on the cross sections for the S_N2 process, but there is little discussion of this in the text.

Author reply:

We thank the reviewer for this valuable comment. The calculations on the modified PES were originally discussed in the *Reaction Dynamics* section, where it is stated: “*When the E2 pathway is completely blocked, S_N2 reactivity undergoes a substantial twofold increase. This observation suggests that in the Cl⁻ + (CH₃)₃CI reaction, the steric hindrance effect corroborates findings from our previous exploration of the strong Lewis base F⁻ reaction: It is not the steric hindrance that hinders the reactivity of S_N2, but the high reactivity of the (CH₃)₃ bulk leads to the predominance of the E2 reaction^{37,42}.*”

To make this clearer, we have now explicitly added “*on a modified PES*” after “*When the E2 pathway is completely blocked*” in the revised manuscript, indicating that these calculations were performed on a modified potential energy surface where the E2 channel was artificially blocked.

7. I thought that the branching ratio data shown in tables 2 and 3 of the SI would be better placed in the main text. In addition, I believe those tables should include the fraction of trajectories which involve the E2 and S_N2 mechanisms. Those numbers appear in the text at various places, and it would be clearer if everything was collected together in one table in the main text.

Author reply:

We thank the reviewer for this valuable suggestion. In the revised manuscript, the data previously shown in Supplementary Tables 2 and 3 have been combined and moved into the main text as the new Table 1, which summarizes both the branching ratios of E2 and S_N2 mechanisms and their trajectory fractions. This integration enhances clarity and allows readers to readily assess the relative contributions of the different pathways directly within the main text.

8. For the non-expert, I thought it would be helpful if the reaction equations (1) appeared earlier in the text, when S_N2 and E2 are first discussed.

Author reply:

We thank the reviewer for this helpful suggestion. In the revised manuscript, the reaction equations (1) have been moved in the text, to **page 3**, where the S_N2 and E2 pathways of the Cl⁻ + (CH₃)₃CI reaction are first introduced. We have also added the sentence: “*Equations (1) summarize the two competing S_N2 and E2 channels for the title reaction*”.

Reviewer #4 (Remarks to the Author):

Review of manuscript titled “Unveiling a flip-over retention mechanism in the gas-phase S_N2

reaction: insights from Cl⁻ + (CH₃)₃CI dynamics” by Xiaoxiao Lu, Jennifer Meyer, Lulu Li, Eduardo Carrascosa, Björn Bastian, Tim Michaelsen, Bina Fu, Dong H. Zhang and Roland Wester.

This manuscript describes theory and experiment on the reaction of Cl⁻ with (CH₃)₃CI forming I⁻ through either a nucleophilic substitution (S_N2) and base-induced elimination (E2) reaction. The E2 reaction forming three fragments I⁻ + HCl + (CH₃)₂CCH₂ and the S_N2 reaction forming two products (CH₃)₃CCl + I⁻. This is distinct from reactions of Cl⁻ + CH₃R (R=halogen) where only the S_N2 reaction is observed. The Wester group has studied many such systems and their experimental expertise for this type of experiment is unquestioned. The reality is that only the I⁻ anion is able to be observed as it is a somewhat unreliable reporter on the details of the reaction due to the fact that it is heavy and slowly moving and the images are of limited velocity resolution. The combination of theory and experiment is quite convincing however. From the I⁻ images one learns that the I⁻ velocities are consistent with E2 elimination

and the analysis indicates that 80% of the reaction proceeds through this channel. This is different from the previously studied F⁻ plus (CH₃)₃CCl system where the E₂ reaction was more dominant and the S_N2 channel was extremely small.

The other more interesting result is the identification of a new reaction mechanism from trajectory calculations on the potential energy surface. Theory on such a large (many atoms and electrons) system is difficult and as I am no expert in the theory I will not comment on this aspect of the manuscript. The new mechanism involves an intriguing spin flip of the t-butyl group that is reminiscent of a sort of hindered roaming of the group such that the I⁻ anion is for a moment bound to a CH₃ group exposing the central c atom to the Cl⁻ anion for reaction. Discovery of a new reaction mechanism is always of fundamental chemical interest. It is not particularly confirmed by the experiment but comes from the theory that the experiment demanded.

In conclusion, if reviewers who are more familiar with the theory feel the predictions of the theory are accurate I would highly recommend publication for this manuscript as it contains a fundamentally new reaction mechanism for a classic reaction.

Author reply:

We sincerely thank the reviewer for the positive and encouraging comments. As detailed in the response to the other reviewers above, the predictions of theory are accurate and important.

Review of manuscript titled “Unveiling a flip-over retention mechanism in the gas-phase SN2 reaction: insights from Cl⁻ + (CH₃)₃CI dynamics” by Xiaoxiao Lu, Jennifer Meyer, Lulu Li, Eduardo Carrascosa, Björn Bastian, Tim Michaelsen, Bina Fu, Dong H. Zhang and Roland Wester.

This manuscript describes theory and experiment on the reaction of Cl⁻ with (CH₃)₃CI forming I⁻ through either a nucleophilic substitution (SN2) and base-induced elimination (E2) reaction. The E2 reaction forming three fragments I⁻ + HCl + (CH₃)₂CCH₂ and the sn2 reaction forming two products (CH₃)₃CCl + I⁻. This is distinct from reactions of Cl⁻ + CH₃R (R=halogen) where only the SN2 reaction is observed. The Wester group has studied many such systems and their experimental expertise for this type of experiment is unquestioned. The reality is that only the I⁻ anion is able to be observed as it is a somewhat unreliable reporter on the details of the reaction due to the fact that it is heavy and slowly moving and the images are of limited velocity resolution. The combination of theory and experiment is quite convincing however. From the I⁻ images one learns that the I⁻ velocities are consistent with E2 elimination and the analysis indicates that 80% of the reaction proceeds through this channel. This is different from the previously studied F⁻ plus (CH₃)₃CCl system where the E2 reaction was more dominant and the SN2 channel was extremely small.

The other more interesting result is the identification of a new reaction mechanism from trajectory calculations on the potential energy surface. Theory on such a large (many atoms and electrons) system is difficult and as I am no expert in the theory I will not comment on this aspect of the manuscript. The new mechanism involves an intriguing spin flip of the t-butyl group that is reminiscent of a sort of hindered roaming of the group such that the I⁻ anion is for a moment bound to a CH₃ group exposing the central c atom to the Cl⁻ anion for reaction. Discovery of a new reaction mechanism is always of fundamental chemical interest. It is not particularly confirmed by the experiment but comes from the theory that the experiment demanded.

In conclusion, if reviewers who are more familiar with the theory feel the predictions of the theory are accurate I would highly recommend publication for this manuscript as it contains a fundamentally new reaction mechanism for a classic reaction.